# Reversible modulation of circadian time with chronophotopharmacology

Dušan Kolarski[1], Carla Miró-Vinyals[1,9], Akiko Sugiyama[2,9], Ashutosh Srivastava [2], Daisuke Ono [3], Yoshiko Nagai[2], Mui Iida[2,4], Kenichiro Itami[2,5], Florence Tama[2,6,7], Wiktor Szymanski[1,8✉], Tsuyoshi Hirota [2,4✉] & Ben L. Feringa [1✉]

The circadian clock controls daily rhythms of physiological processes. The presence of the clock mechanism throughout the body is hampering its local regulation by small molecules. A photoresponsive clock modulator would enable precise and reversible regulation of circadian rhythms using light as a bio-orthogonal external stimulus. Here we show, through judicious molecular design and state-of-the-art photopharmacological tools, the development of a visible light-responsive inhibitor of casein kinase I (CKI) that controls the period and phase of cellular and tissue circadian rhythms in a reversible manner. The dark isomer of photo-switchable inhibitor **9** exhibits almost identical affinity towards the CKIα and CKIδ isoforms, while upon irradiation it becomes more selective towards CKIδ, revealing the higher importance of CKIδ in the period regulation. Our studies enable long-term regulation of CKI activity in cells for multiple days and show the reversible modulation of circadian rhythms with a several hour period and phase change through chronophotopharmacology.

[1] Centre for Systems Chemistry, Stratingh Institute for Chemistry, University of Groningen, Groningen, The Netherlands. [2] Institute of Transformative Bio-Molecules (WPI-ITbM), Nagoya University, Chikusa, Nagoya, Japan. [3] Department of Neuroscience II, Research Institute of Environmental Medicine, Nagoya University, Chikusa, Nagoya, Japan. [4] Division of Biological Science, Graduate School of Science, Nagoya University, Chikusa, Nagoya, Japan. [5] Department of Chemistry, Graduate School of Science, Nagoya University, Chikusa, Nagoya, Japan. [6] Department of Physics, Graduate School of Science, Nagoya University, Chikusa, Nagoya, Japan. [7] Computational Structural Biology Unit, RIKEN-Center for Computational Science, Kobe, Hyogo, Japan. [8] Department of Radiology, Medical Imaging Center, University Medical Center Groningen, University of Groningen, Groningen, The Netherlands. [9] These authors contributed equally: Carla Miró Vinyals, Akiko Sugiyama. ✉email: w.szymanski@umcg.nl; thirota@itbm.nagoya-u.ac.jp; b.l.feringa@rug.nl

Circadian rhythms are endogenous cell-autonomous cycles with a period of ~24 h, which allow organisms to accommodate daily changes. They are considered to be of crucial importance for health through controlling various physiological processes, such as sleep-wake, metabolism, and hormone secretion[1]. Modern lifestyle-caused desynchronization of day–night cycles leads to disturbance of circadian rhythmicity, linked to various diseases and disorders, for instance cardiovascular, gastrointestinal, Alzheimer's, mental diseases, diabetes, cancer, and sleeping sickness[2–7]. Genetics and molecular biology have provided valuable insights in the underlying molecular regulatory mechanisms[8]. Small-molecule clock modulators offer prospects for future chronotherapies through the dose-dependent interaction with circadian regulatory proteins[9]. Whereas high-throughput screening and novel synthetic approaches yielded a variety of molecular modulators of the circadian period[10–14], a key challenge is the lack of spatiotemporal control over circadian rhythms. Due to the presence of the core clock mechanism throughout the body[8], small molecules may affect circadian rhythms not only of the targeted tissue or organs but also elsewhere. Enabling the modulation of the small-molecule potency with light, through the principles of photopharmacology, would result in the possibility of local, non-invasive activation at the desired time and site. In the chronophotopharmacology approach presented here, we address the challenge to remotely and reversibly control the circadian period in cells and tissue with light. In particular, we achieved long-term biological regulation by developing a photoswitchable inhibitor of casein kinase I (CKI) which features drastically enhanced stability and visible light-modulation of the inhibitory effect.

In mammals, circadian oscillations are driven by negative feedback loops in which the CLOCK-BMAL1 transcription factors bind to the E-box elements on the gene promoters, and activate the transcription of Period (Per) and Cryptochrome (Cry) genes (Fig. 1a)[8]. The PER protein is phosphorylated by a family of CKI proteins (isoforms δ, ε, and α) promoting proteasomal degradation and keeping the circadian rhythms on the 24 h base[8]. It has been shown that deceleration of this process by CKI inhibition leads to the circadian period lengthening[10,15,16]. The purine-based CKI inhibitor – longdaysin (1, Fig. 1b) acts as circadian period modulator with one of the strongest known period lengthening effects in a variety of cells and tissues[10].

Here, we use longdaysin as a starting point in the photopharmacological approach[17,18], aiming to create a photoswitch-modified inhibitor that can be reversibly converted with light between isomers of distinctly different CKI inhibitory potency (Fig. 1a). Such longdaysin analogs (2, Fig. 1b) will ultimately allow precise time- and site-selective action. Toward this goal, we introduce into the structure of longdaysin an azobenzene moiety, a member of a common class of photoswitches that upon irradiation undergo a conversion from the trans (thermally stable) to the cis (thermally unstable, reverts back to trans in time) isomer, while the reverse cis-to-trans process is both photochemically and thermally driven (Fig. 1b). With these photoswitchable CKI inhibitors in hand, a reversible modulation of the circadian period in cells and tissue is achieved. Additionally, meticulously optimized visible light-responsive modulator 9 allows for a transient period change during the cellular assay that results in the phase shift.

## Results and Discussion
### Choosing a optimal scaffold for a photoswitchable circadian rhythm modulator.
The development of azobenzene-modified longdaysin variants requires several parameters to be established, including a large difference in biological activity between the photoisomers (while retaining the overall high potency of the original bioactive molecule). Also, particularly demanding are extended half-lives of the cis-isomer to allow its evaluation in long-term (several days) bioactivity assays, (photo)chemical stability and the use of visible light for the isomerization in both directions to enable non-invasive operation in vitro and in cellulo.

First, we aimed to establish the optimal molecular architecture of photocontrolled derivatives 2 (Fig. 1b). To minimize structural and electronic changes introduced upon modification of longdaysin with azobenzene, a phenyldiazenyl group was introduced instead of the $CF_3$ moiety, retaining hydrophobic interactions with CKI without compromising potency. Given that a structure-activity relationship (SAR) study on the benzylamine moiety of longdaysin is not available, our initial optimization included all three possible azobenzene regioisomers (3-5, Fig. 2a) and the effect of the structural modification on the photochemical and biological activity was investigated (see Supplementary Information for details of synthesis and characterization).

Before irradiation, all compound solutions were thermally adapted and contained pure trans-isomers (>98%), while irradiation at $\lambda_{max} = 365$ nm resulted in formation of 69–93% of the cis-isomer (Fig. 2b). Using visible light ($\lambda_{max} = 450$ nm or white light) for back-switching, a trans-enriched photostationary state (PSS) was recovered, containing 68–74% of the trans-isomer (Supplementary Figs. 1–3). No significant fatigue upon six switching cycles, nor any reduction or light-induced oxidation of the azo group under enzymatic and cellular assay-mimicking conditions was observed (Supplementary Figs. 8–10). All modulators showed good solubility in aqueous media (Supplementary Table 1). The half-lives of the thermally unstable cis-3-5 are over 3 h in the kinase assay buffer, and even longer in the cell culture medium (>24 h, 11 h, and >24 h for 3, 4, and 5, respectively, Fig. 2b and Supplementary Figs. 15–17), rendering them suitable for the in vitro CKI assays in which the potency of both trans- and cis-enriched samples are evaluated to identify the best substitution pattern.

In the CKIα activity assay (for experimental details see Supplementary Information), para-substituted modulator 3 showed no obvious light-dependent changes of CKIα inhibition ($IC_{50, \, dark} = 3.6$ μM and $IC_{50, \, light} = 4.5$ μM, Fig. 2c) while meta-substituted photoswitch 4 exhibited a substantial light-induced inactivation ($IC_{50, \, dark} = 1.9$ μM and $IC_{50, \, light} = 8.6$ μM, Fig. 2c), constituting a 4.5-fold decrease in activity under irradiation. Control experiments showed a negligible effect of UV irradiation on the potency of longdaysin ($IC_{50, \, dark} = 5.6$ μM, $IC_{50, \, light} = 4.3$ μM, Supplementary Fig. 23). In contrast, with ortho-substituted modulator 5, the opposite influence of light on the kinase assay outcome was observed. Even though full CKIα inhibition could not be achieved due to solubility issues at high concentrations, cis-enriched-5 showed a 1.8-fold stronger inhibition than the trans-isomer ($IC_{50, \, dark} = 96$ μM and $IC_{50, \, light} = 54$ μM, Fig. 2c). Molecular docking of each inhibitor with CKI was performed, which supported the experimental data (see Supplementary Information for details, Supplementary Fig. 38).

Next, we tested if the observed light-dependent CKIα inhibition translates into photomodulation of the circadian period in human U2OS cells (Fig. 2d). The Bmal1-dLuc reporter cell line contains the coding sequence of destabilized luciferase driven by the Bmal1 gene promoter sequence[19]. The cell culture medium for luminescence recording includes a high concentration of luciferin (0.2 mM), a substrate for luciferase, which strongly absorbs UV light (Supplementary Fig. 30). In control cells, luminescence rhythms show a period of ~24 h, whereas the inhibition of CKI causes period lengthening due to slowing down the phosphorylation of PER. As the circadian period change is directly correlated with the potency and concentration of the

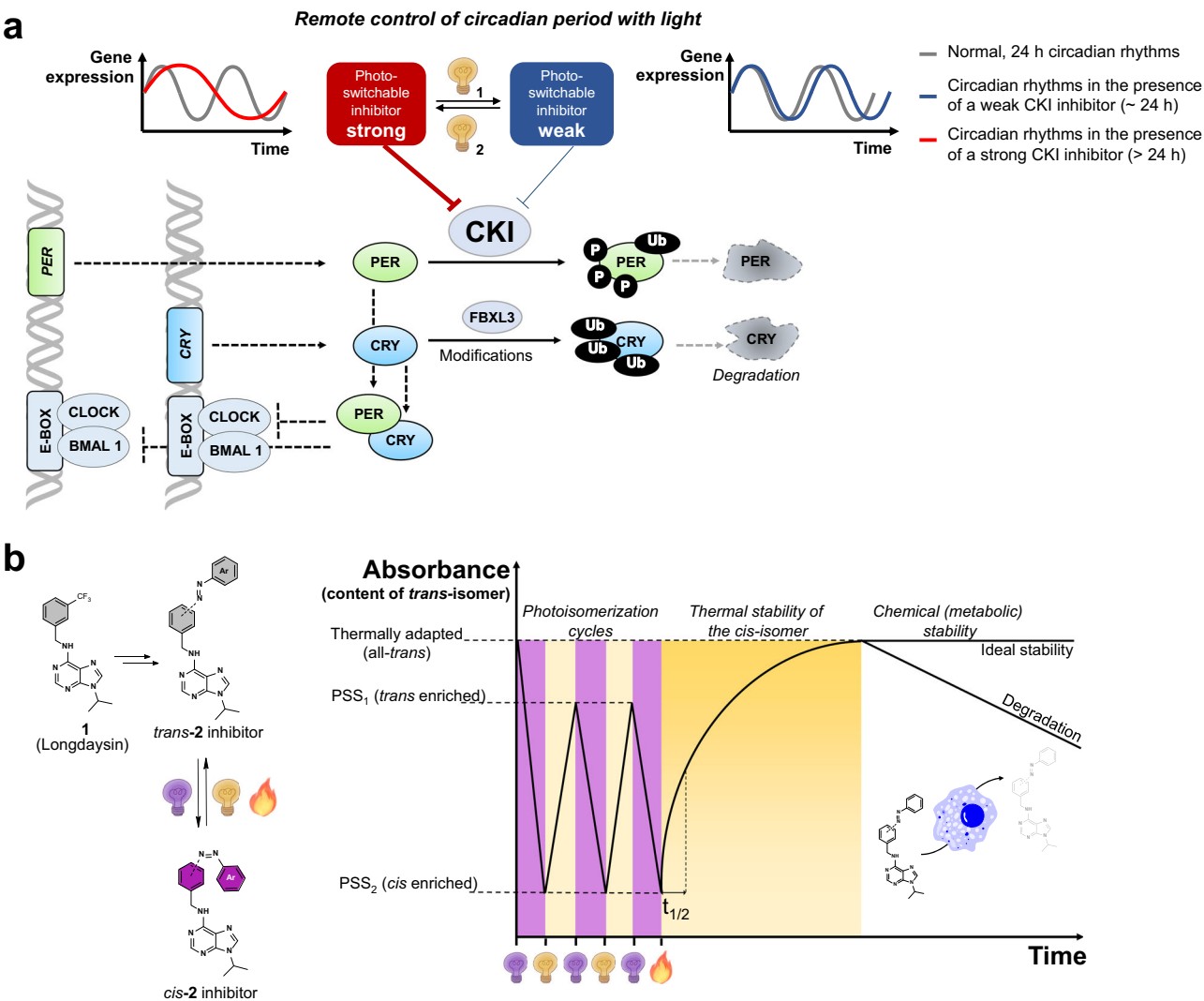

**Fig. 1 The concept of the reversible chronophotopharmacology. a** Schematic representation of the circadian clock transcription-translation autoregulatory feedback loop, and its photo-modulation by photoswitchable inhibitor of CKI. Both isomers of the photoswitchable bioactive molecule show inhibition of CKI, but the affinity can be modulated with light, with one photoisomer (shown in red) being a stronger inhibitor than the other (blue). **b** Key parameters of the photoswitchable circadian modulators (**2**), based on longdaysin (**1**) structure, that must be considered for a successful application in reversible modulation of the circadian rhythm. *Trans*-to-*cis* isomerization can be performed by irradiation with one wavelength (purple light), switching the thermally adapted *trans*-isomer (all-*trans*) to the *cis*-enriched isomer (PSS$_2$). Back-isomerization can be achieved by another wavelength (white light) yielding the *trans*-enriched isomer distribution (PSS$_1$). Photochemical stability is measured by repetitive isomerization at two different wavelengths (photoisomerization cycles), and if there is no fatigue present, PSS$_1$ and PSS$_2$ will be reached each time. If left in the dark, the *cis*-isomer will thermally isomerize back to more stable *trans*-isomer (thermal stability of the *cis*-isomer) in a first order process. The time needed for half the *cis*-isomer to isomerize back to the *trans* is called half-life (t$_{1/2}$). Additionally, due to lengthy assays to measure circadian rhythms (5–6 days), azobenzene modulators should possess metabolic stability to maintain light-dependent circadian period modulation.

inhibitor[10], the cells were treated with photoswitchable modulators using a broad range of concentrations.

In comparison to the kinase assay, the direct use of UV light for preventing a thermal back-isomerization of the *cis*-isomer was not feasible due to the cell-damaging effect and the presence of luciferin in the medium. Therefore, the modulators were kept in the dark or photo-isomerized in DMSO prior to the addition, without further irradiation during the circadian assay. All three modulators lengthened the period in the U2OS cells both in the dark and upon irradiation (Fig. 2d). The relative period lengthening in cells corresponds to their potencies obtained in the CKIα enzymatic assay: **3** showed 4 h period lengthening at 24 μM but no light-modulation, **4** exhibited a minor light-induced

deactivation (ca. 2 h) at 3 μM, while **5** showed a weak activation at 72 μM (Fig. 2d). Despite a pronounced 4.5-fold difference in potency between *trans*-**4** and *cis*-**4** observed in the enzymatic assay, the weak difference observed in the cellular assay can be attributed to the relatively short half-life of the *cis*-isomer (Fig. 2b). While **5** showed only 1.8-fold better CKIα inhibition upon irradiation, and only a 69:31 *cis:trans* ratio upon isomerization, the effect of irradiation was still noticeable in the cells at the concentration of 72 μM (Fig. 2d). Altogether, based on the efficient generation of the *cis*-isomer under irradiation, retained potency and promising light-modulation of CKIα inhibition, the *meta*-substituted modulator **4** was identified as representing lead substitution pattern for further investigation.

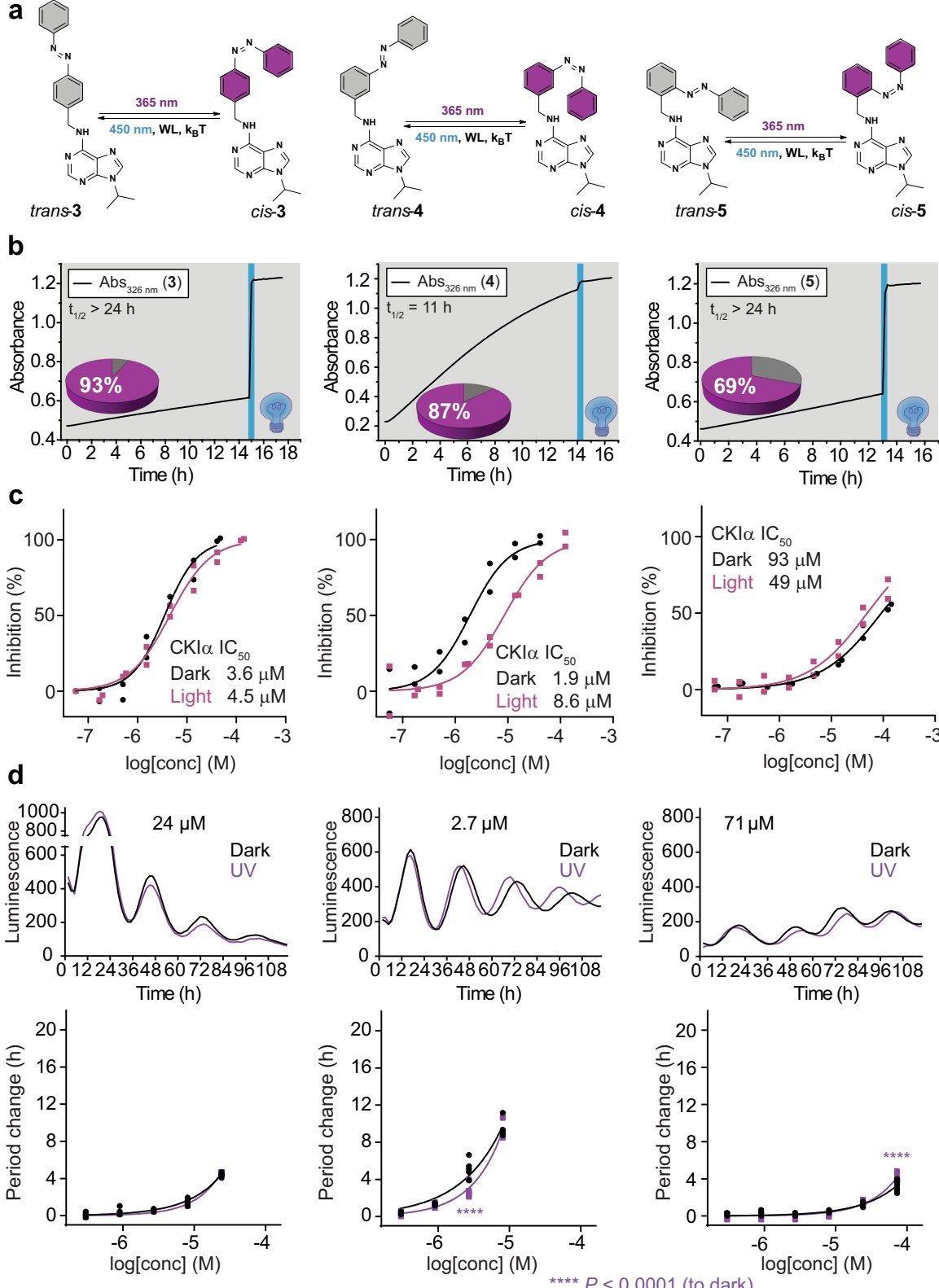

**Enhancement of the circadian rhythm photomodulation effect**. The key optimization points to achieve maximum potency difference between the irradiated and non-irradiated sample are the amount of *cis*-isomer generated upon irradiation and its half-life. To evaluate the relative importance and interplay between these factors, we designed and tested azobenzene-modified longdaysin analogs **6-8** (Fig. 3a), in which different substituents were introduced in the most electronically and sterically sensitive positions (*ortho* and *para*) to the azo-group (Fig. 3a). The presence of a *para*-methoxy substituent (compound **6**) might enable high switching ratios in both directions[20], while substituting two *ortho* positions (compound **7**) usually yields photoswitches with extended half-lives[21]. Furthermore, the effect of a twisted *cis*-form of the *ortho*-dimethyl substituted arylazopyrazoles (AAPs,

**Fig. 2 Photochemical and biological evaluation of modulators 3-5. a** Structures and photoisomerization scheme of modulators **3**-**5**. **b** *Cis*-to-*trans* thermal isomerization in cell culture medium followed in the dark. Photostationary states (PSSs, pie charts) were reached upon irradiation of DMSO solution (2 mM, 25 °C) with UV light ($\lambda_{max} = 365$ nm) and subsequent dilution in the cell culture medium to obtain the final concentration of 40 µM. The amount of the *cis*-isomer is shown in pie chart (purple). After 12–14 h in the dark (gray background), blue light ($\lambda_{max} = 450$ nm, blue rectangle) was applied for 8 min to confirm the presence of the remaining *cis*-isomer and provided the *cis*-to-*trans* ratio. **c** Photo-modulation of the CKIα inhibition, shown as the dose-response curve for the inhibitor kept in the dark (black line) and irradiated with UV light (purple line) before addition to the enzymatic reaction mixture (30 min) and during the assay (3 h). **d** Luminescence rhythm profiles (average of $n = 6$) and period-changes of U2OS reporter cells relative to DMSO control, applied with the compounds kept in the dark (black lines) or irradiated with UV light (purple line). Data for longdaysin and DMSO-treated cells are shown in Supplementary Fig. 24a, b, respectively. Luminescence is given in arbitrary units. $n = 2$ biologically independent samples for the enzymatic assay (**c**) and $n = 6$ biologically independent samples for the cellular assay (**d**). Two-way ANOVA followed by a Sidak's multiple comparisons test was used for statistical analysis. *P* value is shown in the figure.

compound **8**) was explored to increase the difference in binding affinity between the two isomers and their PSS distributions to enhance an overall biological effect[22].

Modulators **6**-**8** demonstrated chemical and photochemical stability in aqueous media and sufficiently long half-lives (>3 h) for the enzymatic activity assay (Supplementary Figs. 4–6, 11–13, and 34–36). *para*-Methoxy–substituted analog **6** showed increased solubility and allowed the *cis*-isomer to be almost quantitatively (97%) obtained upon UV light irradiation (Fig. 3a). However, the short half-life of 18 h in cell culture medium still poses a drawback of this modulator. The *cis*-isomer was stabilized further by two *ortho*-fluorine atoms in compound **7**, taking advantage of the σ-electron-withdrawing effect ($t_{1/2} > 50$ h, 62% *cis* isomer under irradiation; Fig. 3a)[21]. Despite the fact that the electron-poor azo-group is generally more susceptible to reduction[23], modulator **7** exhibited both chemical and photo-chemical stability (Supplementary Figs. 12 and 35). Its solubility was similar to the parent modulator **4**, even though two highly lipophilic atoms were introduced (Fig. 3a). Heteroaromatic photoswitch **8** showed low fatigue, no observable reduction or oxidation, high PSS distribution (97% *cis* isomer under irradiation) with a moderately long half-life ($t_{1/2} = 26$ h; Fig. 3a, and Supplementary Figs. 6, 13, 20, and 36).

Photoresponsive modulators **6**-**8** were evaluated in the enzymatic and cellular assays (Fig. 3b, c). All modulators retained potency and exhibited a light-induced decrease in inhibition of CKIα similarly to modulator **4**. The increased PSS distribution in the case of **6** and **8** yielded photoswitchable kinase inhibitors with the most pronounced photo-modulation, showing 5-and 8-fold differences in activity, respectively. Despite a moderate PSS distribution, *cis*-enriched **7** was four times less potent than the *trans*-isomer. Subsequently, light-dependent modulation of the cellular circadian rhythms was examined, and revealed highly pronounced reversible modulation of the circadian period, using modulators **7** and **8**. Both photoswitches under dark conditions have a strong period lengthening effect, which can be suppressed by UV light irradiation (black and purple lines, Fig. 3c). Furthermore, a partial recovery of the biological activity for **7** and full recovery in case of **8** was achieved upon back-isomerization with white light (red lines, Fig. 3c). Remarkably, modulator **7** showed the most pronounced period lengthening deactivation, even though the light-induced activity difference in the kinase assay was lower than modulators **6** and **8**. Comparing the half-lives of these three modulators (Fig. 3a), the results clearly show that a long half-life (c.f. compound **7**) is crucial for obtaining a successful photo-modulation of the circadian period and that this parameter is more important than the high PSS distribution (compounds **6** and **8**). Having established this important design criterion, we focused on minimizing the background activity of the non-desired *trans*-isomer and enabling visible-light operation.

**Visible light modulation of the circadian period.** Inspired by the finding by Hecht et al.[24], that introduction of four *ortho*-fluoro atoms as σ-electron-withdrawing groups leads to a good n → π* band separation (~50 nm) of azobenzene isomers, and thus enables reaching high PSS distributions for both isomers under visible light irradiation, we synthesized *ortho*-tetra-fluoroazobenzene derivative **9** (Fig. 4a). It exhibits a highly thermally stable *cis*-form ($t_{1/2} > 50$ h in cell culture medium) and shows efficient photoisomerization in both directions (Fig. 4b, c). Specifically, the use of green light ($\lambda_{max} = 530$ nm) led to efficient isomerization of modulator **9** by excitation of the n → π* absorption band of the *trans*-isomer, while the reverse *cis*-to-*trans* isomerization was achieved with violet light ($\lambda_{max} = 400$ nm, Fig. 4a and Supplementary Fig. 22). Repeated illumination cycles revealed very low fatigue (Fig. 4c). Besides allowing the switch to be operated with visible light, without concerns for toxicity and very limited tissue penetration depth characteristic of UV light, it also enabled the reversible photoisomerization in the cell culture medium containing luciferin (Fig. 4b and Supplementary Fig. 37).

For modulation of the CKI activity, isomers of compound **9** were prepared by thermal adaptation (>99% *trans*) or by photoisomerization with green light (86% *cis*). Photoisomerization was conducted prior to application, since, due to very long half-life of the *cis* form, no irradiation during the course of the assay was required. The potency of **9** was comparable to that of longdaysin. The obtained $IC_{50}$ values of CKIα inhibition again indicate better inhibition in the *trans*-form (Fig. 4d), but this time with only 1.5-fold difference in respect to the *cis*-isomer. The effect of reversible photomodulation of the circadian period was observed in U2OS cells (Fig. 4e), and notably the effect was more pronounced. The *trans*-isomer showed a strong, 4 h period lengthening at the concentration of 8 µM (black, Fig. 4e), while upon photoisomerization to the *cis*-isomer with green light in vitro, the period lengthening was almost fully suppressed, giving a 1-h period change at the same concentration (green, Fig. 4e). The remaining circadian period lengthening can be attributed to activity of *cis*-isomer (Fig. 4d), as well as leftover *trans*-isomer (≥14%, Fig. 4c), being the result of non-quantitative photoisomerization and a very slow but still relevant thermal relaxation of the *cis*-isomer during the course of the assay (Fig. 4b). Remarkably, in situ back-photoisomerization in U2OS cells with violet light fully restored the period lengthening of the *trans*-isomer, resulting in a 4-h lengthened circadian period (violet, Fig. 4e). Treatment of the dark samples with only violet light had no effect on the compound activity on the circadian period (gray, Fig. 4e).

Aiming to explain the strong inactivation of the compound observed upon irradiation in the cellular assay (vide infra, Fig. 4e), which is more pronounced than that observed in the CKIα assay, we proceeded with testing the photo-modulation of the activity of the CKIδ isoform (Fig. 4d). Remarkably, it was found that $IC_{50}$

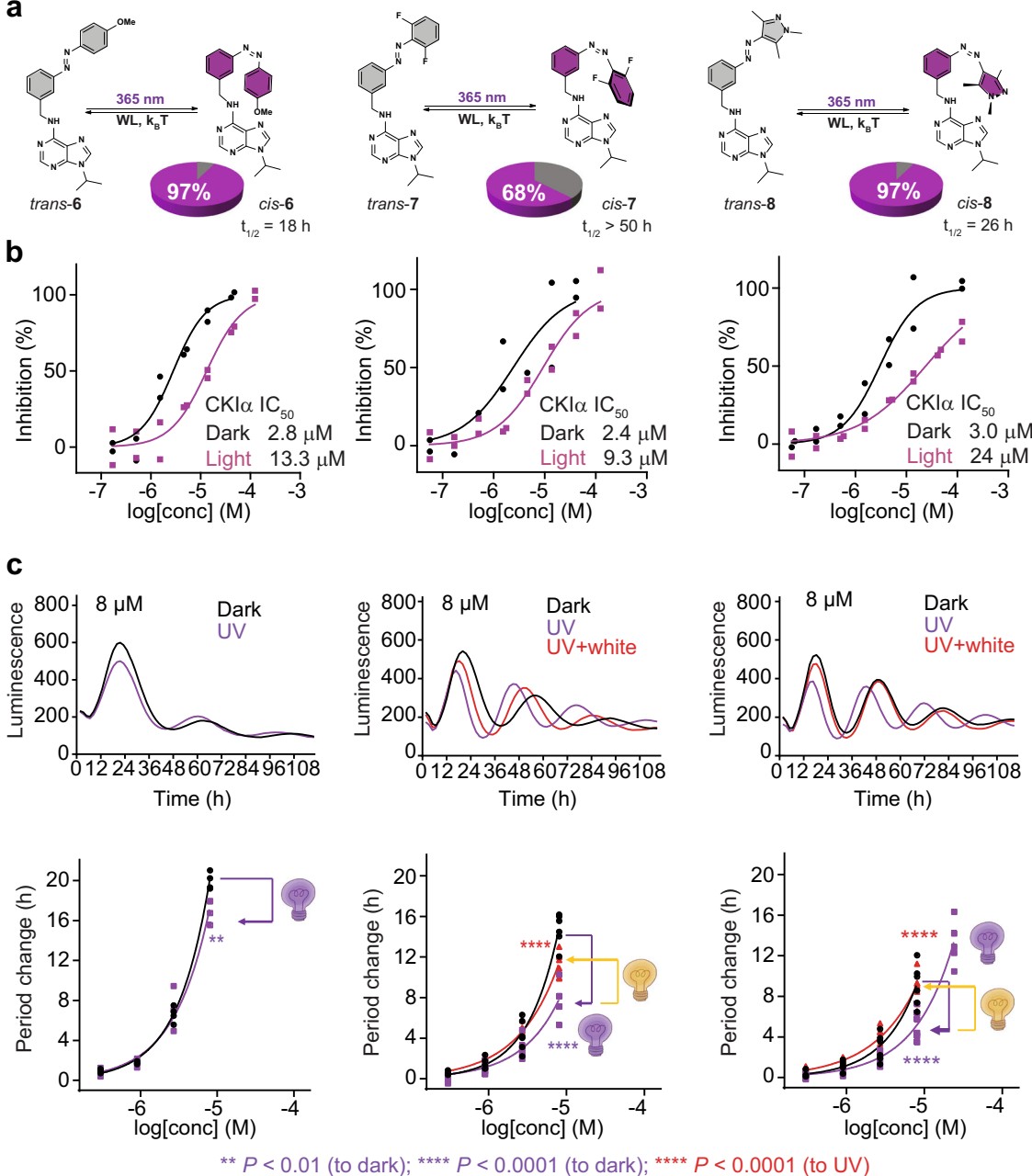

** *P* < 0.01 (to dark); **** *P* < 0.0001 (to dark); **** *P* < 0.0001 (to UV)

**Fig. 3 Photochemical and biological evaluation of modulators 6-8. a** Chemical structures of **6-8**, photochemical *trans*-to-*cis* isomerization (UV light, $\lambda_{max}$ = 365 nm), thermal- or white- light- induced back-isomerization and corresponding percentage of *cis* isomer (pie chart, purple) that can be generated under irradiation at PSS and thermal half-lives of modifiers **6**, **7**, and **8** in cell culture medium (40 µM, 35 °C). Upon reaching PSS distribution under UV light irradiation, thermal back-isomerization was followed in dark for 12–14 h. **b** Kinase (CKIα) and **c** cellular (U2OS cells) assay data for modulators **6-8**. Dose-response curves of the kinase inhibition or the circadian period lengthening under the dark condition are shown in black, upon irradiation with UV light in purple, and back-switching with white light is indicated with red lines. Effects of dark, UV light (purple bulb), and UV light followed by white light (yellow bulb) conditions on luminescence rhythm profiles (average of *n* = 6) are also shown in the top panels of **c** (8 µM of each inhibitor). Data for longdaysin- and DMSO-treated cells are shown in Supplementary Fig. 24c, d, respectively. Luminescence is given in arbitrary units. *n* = 2 biologically independent samples for the enzymatic assay (**b**), and *n* = 4–6 biologically independent samples for the cellular assay (**c**). Two-way ANOVA followed by a Sidak's multiple comparisons test was used for statistical analysis. *P* value is shown in the figure.

for the *cis*-isomer is 3.7-times higher than for the *trans*-isomer. This constitutes the first case of an inhibitor that has the same potency for the two kinase isoforms in its dark state (*trans*-**9**) while upon irradiation, the other isomer (*cis*-**9**) shows profoundly different affinity for the isoforms. This unique feature allows to differently modulate the inhibition of two kinase isoforms with

the same small molecule, which provides a tool to elucidate the function of the key proteins involved in circadian rhythm regulation. To further explain the difference in the CKI-isoform affinity, we used molecular docking to investigate and compare the binding energy of *trans*-**9** and *cis*-**9** in complex with CK1δ with that of CK1α (Supplementary Fig. 38b). The binding energy

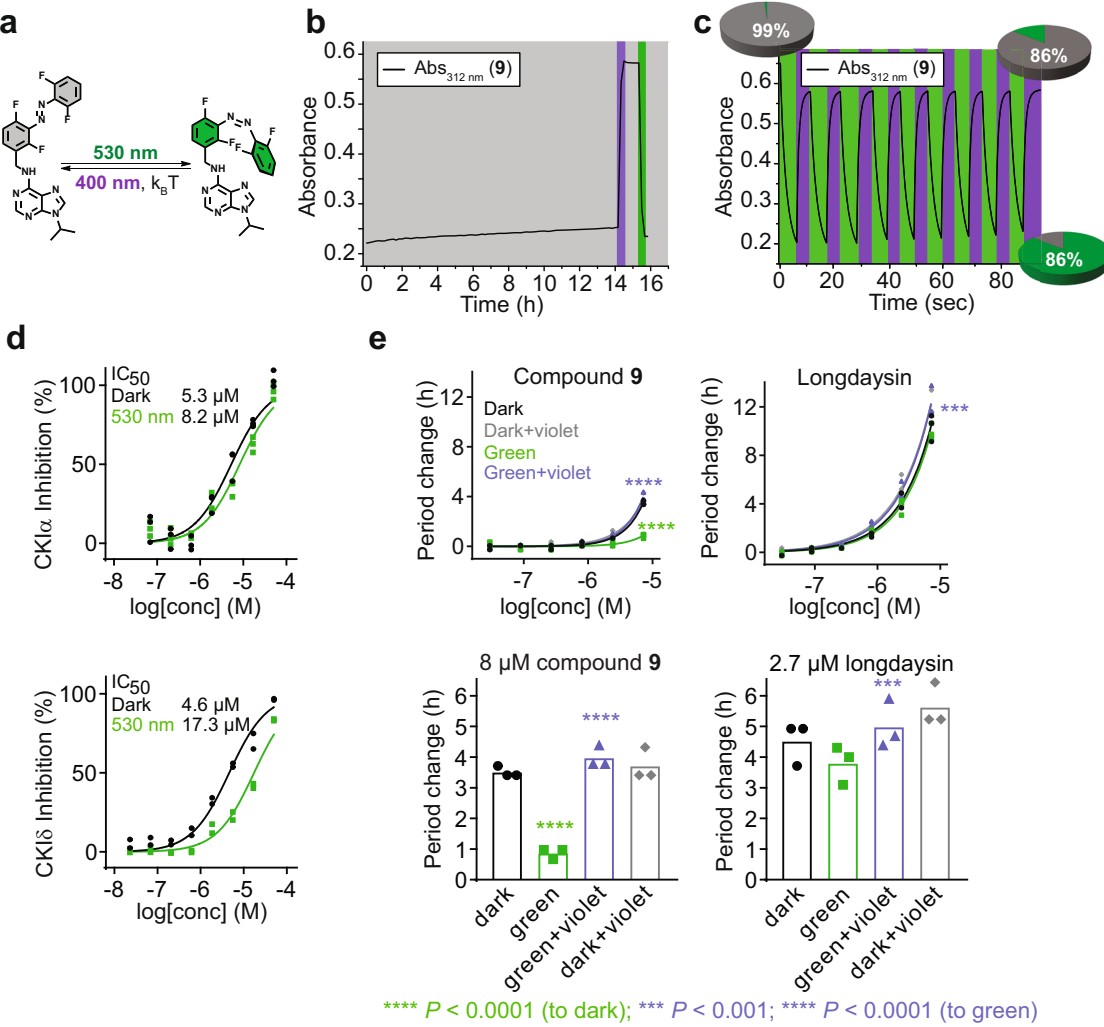

**Fig. 4 Photochemical and biological evaluation of modulator 9. a** Photoisomerization scheme using green light ($\lambda_{max}$ = 530 nm) for *trans*-to-*cis* and violet light ($\lambda_{max}$ = 400 nm) for back-isomerization. **b** Thermal half-live of modifier **9** in cell culture medium ($t_{1/2}$ > 50 h, 35 °C, 40 μM). Upon reaching PSS distribution with green light, thermal back-isomerization was followed in the dark for 14 h (gray background). After this period, violet light was applied (violet rectangle) for 8 min to confirm the presence of the remaining *cis*-isomer and estimate *cis*-to-*trans* ratio. Green light allowed for switching to the *cis*-isomer once again (green rectangle), showing that in situ isomerization is possible. **c** Reversible photochromism in cell culture medium followed at 312 nm absorbance. Almost no fatigue was observed after nine repetitions (cell culture medium, 40 μM, 35 °C) of irradiation with violet and green light. **d** Light-modulation of the CKIα and CKIδ inhibition. The *trans*-isomer (dark, shown in black) shows higher potency than the *cis*-enriched mixture (green light, shown in green). **e** Circadian period modulation in U2OS cells with light. The thermally adapted sample is shown in black (dark), irradiated with green light in vitro (pre-incubation, shown in green), with violet light in cellulo (post-incubation, shown in gray), and with green light in vitro followed by violet light in cellulo in purple. The control experiment with longdaysin is also shown. Luminescence rhythm profiles and data for DMSO-treated cells are shown in Supplementary Figs. 25 and 26, respectively. Luminescence is given in arbitrary units. $n$ = 3 biologically independent samples for the CKIα enzymatic assay (**d**), $n$ = 2 biologically independent samples for the CKIδ enzymatic assay (**d**), and $n$ = 3 biologically independent samples for the cellular assay (**e**). Two-way ANOVA followed by a Sidak's multiple comparisons test (**e**, top panels) or one-way ANOVA followed by a Tukey's multiple comparisons test (**e**, bottom panels) was used for statistical analysis. $P$ value is shown in the figure.

for *trans*-**9** was lower than for *cis*-**9** in case of CK1δ complex (Supplementary Fig. 38b), corroborating the IC$_{50}$ values determined for these compounds in vitro (Fig. 4d).

With an optimized, visible light-responsive and thermally stable modulator **9**, we tested its long-term metabolic stability in cells, and the possibility to modulate the pace of biological clock 3 days after its application to the cells (Fig. 5). A thermally adapted sample (99% *trans*-**9**) was applied to cells and the circadian period change was monitored for 3 days in the dark (Fig. 5a). The cells treated with *trans*-**9** at 8 μM exhibited a period change of 5 h (green and black lines, Fig. 5a, gray background). Then, half of the plate containing cells was irradiated with green light (provoking *trans*-to-*cis* isomerization and lowering the

inhibitory effect) and the circadian period was monitored for three additional days. After this treatment, non-irradiated cells displayed an even longer circadian period (10 h longer period than in DMSO control at 8 μM, black line, Fig. 5a, green background) while irradiated cells revealed a major suppression of the period lengthening, reaching only 2 h lengthening at 8 μM (green line). These results clearly demonstrate that modulator **9** can reversibly alter the circadian period in long-term experiments.

To confirm that modulator **9** can switch the circadian phenotype in both directions, additional experiments were performed; the cells were initially irradiated with green light or kept in the dark. Monitoring rhythms of the green light-irradiated

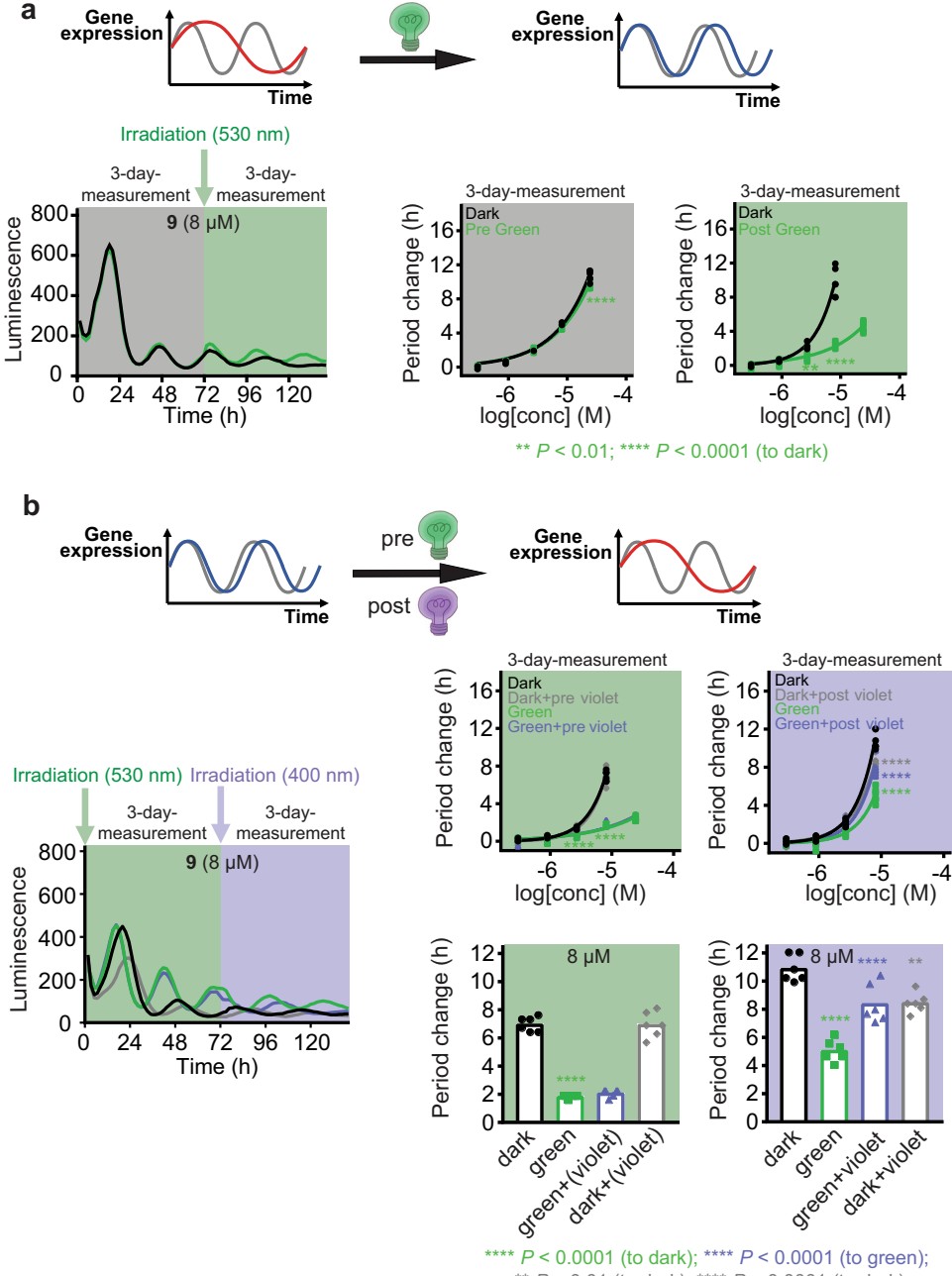

**Fig. 5 Biological evaluation of compound 9 for a long-term photo-modulation of the circadian period. a** The circadian period change over 6 days of the U2OS cellular assay starting with *trans*-**9**. Thermally adapted **9** was applied to the cells, and the period change was monitored in the dark for 3 days (gray background, black and green lines). Half of the plate was kept in dark for additional three days (black line) while the other half was irradiated shortly with green light ($\lambda_{max} = 530$ nm), followed by 3 days recording in dark (green background, green line). **b** The circadian period change over six days of the cellular assay starting with *cis*-enriched or *trans*-**9**. Three-day period change was monitored in the dark upon the application of thermally adapted **9** (black and gray lines) or following short green light irradiation in cellulo (green background, green and violet lines). On the 3rd day, half of the cells with *cis*-enriched **9** were irradiated shortly with violet light ($\lambda_{max} = 400$ nm, violet background, violet line) and half of the cells with *trans*-**9** were irradiated shortly with violet light (gray line). The other half of the cells with *cis*-enriched and *trans*-**9** were kept in dark and recorded additional 3 days (green and black lines, respectively). Luminescence is given in arbitrary units. $n = 6$ biologically independent samples. Two-way ANOVA followed by a Sidak's multiple comparisons test (**a**, **b**, top panels) or one-way ANOVA followed by a Tukey's multiple comparisons test (**b**, bottom panels) was used for statistical analysis. *P* value is shown in the figure. Data for longdaysin and DMSO-treated cells are shown in Supplementary Fig. 27.

cells exhibited a suppressed period change (2 h at 8 μM, green and violet line, Fig. 5b, green background) in comparison to the sample kept in dark (7 h at 8 μM, black and gray line). In situ irradiation of U2OS cells with violet light after the third day of monitoring induced *cis*-to-*trans* isomerization and successfully enhanced the circadian period lengthening to 8 h at 8 μM (violet

line, Fig. 5b, violet background), which was larger than dark treatment of green-irradiated cells (5 h, green line) and similar to violet light treatment of dark-kept cells (8 h, gray line). These data show that applying switchable small molecule inhibitor **9** to cells allows in situ photochemical suppression (*cis*-enriched **9**) and enhancement (*trans*-enriched **9**) of the circadian period

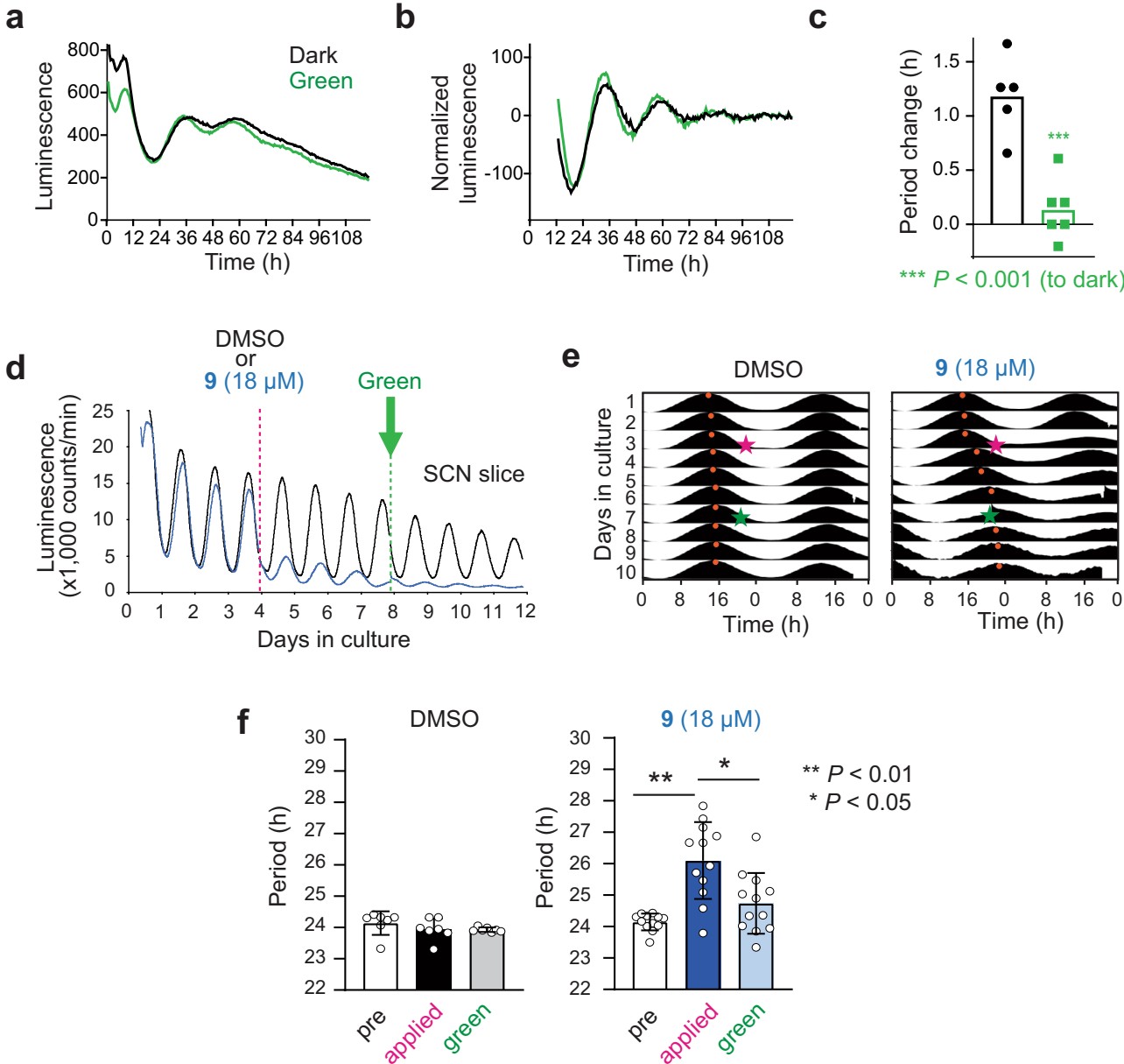

**Fig. 6 Ex vivo photo-modulation of the circadian period in mouse tissue explants. a–c** The spleen explants of the *Per2::Luc* knock-in reporter mice were applied with 12 μM compound **9** and irradiated with green light ($\lambda_{max} = 530$ nm) for 0 min (dark, black) or 30 min (green). Luminescence rhythms were then monitored and shown in **a** (mean of $n = 5$ for dark and $n = 6$ for green light). Baseline-subtracted data are shown in **b**. Period changes compared to a DMSO control are plotted in **c**. Data for DMSO-treated tissues are shown in Supplementary Fig. 28. Luminescence is given in arbitrary units. $n = 5$ biologically independent samples for dark and $n = 6$ for green light. Two-sided Student's *t* test was used for statistical analysis. *P* value is shown in the figure. **d–f** The SCN explants of the *Per2::Luc* knock-in reporter mice were applied with DMSO or 18 μM compound **9** on day 4 (pink) and irradiated with green light on day 8 (green). Luminescence rhythms of a representative sample for each condition are shown in **d**. Baseline-subtracted and normalized data of **d** are shown in **e**. Peaks are indicated by orange dots. Period changes are plotted in **f**. Data are mean with SD ($n = 7$ biologically independent samples for DMSO and $n = 12$ for compound **9**). One-sided Friedman test with post hoc Steel-Dwass test was used for statistical analysis. *P* value is shown in the figure.

lengthening activities even under long-term irradiation conditions. The control experiment using longdaysin as the circadian modulator showed a negligible influence of green or violet light on the period change (Supplementary Fig. 27).

Next, we applied the light-dependent period control to the tissue level using explants of *Per2::Luc* knock-in reporter mice. The mice express PER2-luciferase fusion protein under control of the endogenous *Per2* promoter[25]. According to rhythmic expression of PER2 proteins, luminescence intensity shows circadian changes. The spleen explants were treated with thermally adapted *trans*-**9**,

then irradiated with green light ($\lambda_{max} = 530$ nm) or kept in dark, and luminescence rhythms were measured (Fig. 6a, b). Consistent with the cellular assay results, *trans*-**9** in dark caused period lengthening compared with DMSO control, and green light irradiation suppressed the period-lengthening effect (Fig. 6c). In addition to the peripheral clock in spleen, we investigated the effect of compound **9** on the central clock in the hypothalamic suprachiasmatic nucleus (SCN) that controls behavioral rhythms. Thermally adapted *trans*-**9** induced period lengthening, and its effect was reduced upon green light irradiation (Fig. 6d–f).

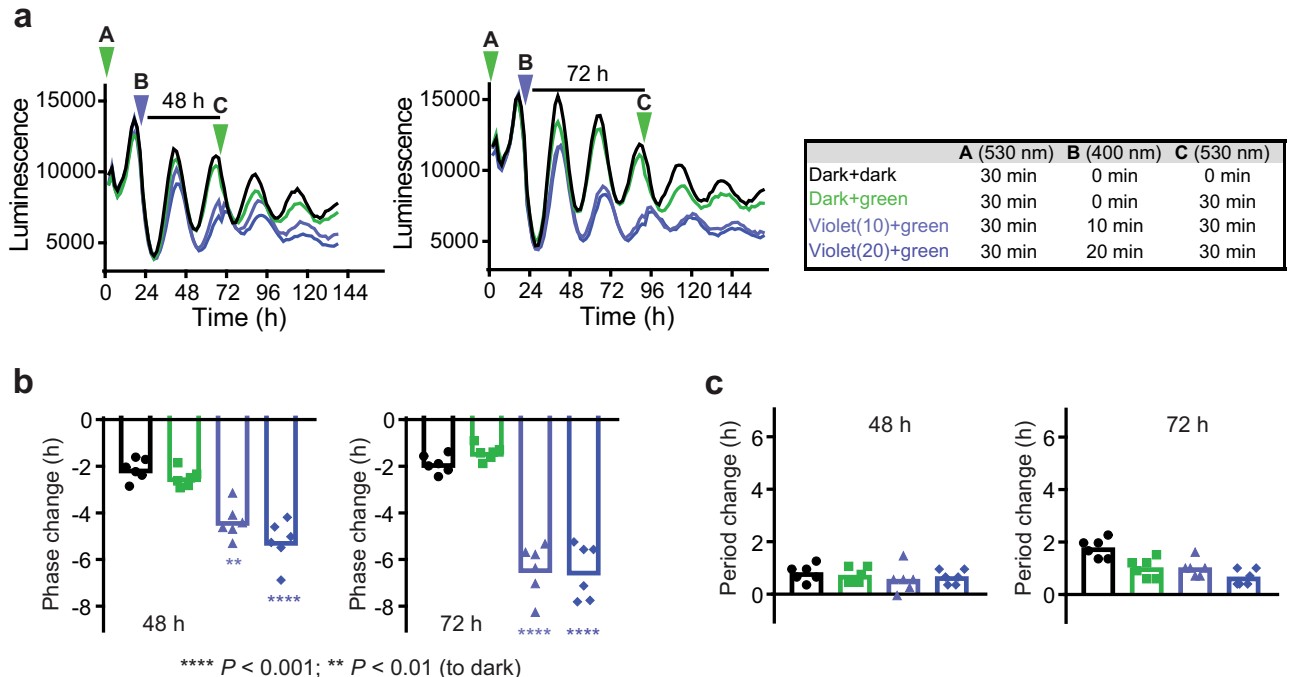

**Fig. 7 Circadian phase control using chronophotopharmacology.** Green light ($\lambda_{max}$ = 530 nm)-irradiated compound **9** was applied to U2OS cells at 0 h (A). On the next day (B), the cells were irradiated with violet light ($\lambda_{max}$ = 400 nm) for 0 min (black and green lines), 10 min (light violet line), or 20 min (dark violet line). After 48 h or 72 h (C), the cells were irradiated with green light ($\lambda_{max}$ = 530 nm) for 0 min (black line) or 30 min (green, light violet, and dark violet lines). Luminescence rhythm profiles (average of $n$ = 6) are shown in **a**. Phase and period changes relative to DMSO controls after time point C are shown in **b** and **c**, respectively. Data for DMSO-treated cells are shown in Supplementary Fig. 29. Luminescence is given in arbitrary units. $n$ = 6 biologically independent samples. One-way ANOVA followed by a Tukey's multiple comparisons test was used for statistical analysis. $P$ value is shown in the figure.

Together, compound **9** enabled light-dependent inducible control of the circadian period at both cellular and tissue levels.

**Modulating the circadian phase with light**. Hypothetically, a transient change in period will lead to a phase change, consequently providing a relevant approach to jetlag treatment. Therefore, we applied visible-light responsive modulator **9** to control the circadian phase (Fig. 7). The compound was initially irradiated with green light for inactivation (*cis*-enriched **9**, Fig. 7a, timepoint A) and applied to U2OS cells for luminescence rhythm measurement. On the next day, the cells were irradiated with violet light for compound activation or kept in dark (Fig. 7a, timepoint B). The transient period lengthening by *trans*-enriched **9** was reversed by green light irradiation after 48 or 72 h (Fig. 7a, timepoint C), which resulted in 2 h (48 h condition) or 4 h (72 h condition) phase delay compared with dark samples (Fig. 7b), with period returned to normal (Fig. 7c). The results show the power of chronophotopharmacological approach not only for period but also for phase regulation of circadian rhythms, providing a proof-of-concept for a potential new therapy of circadian misalignments.

Altogether, we realized the development of photocontrolled modulators of circadian rhythm. Meticulous evaluation of photochemical and biological properties of the molecules **3-8**, led to the development of compound **9** as a visible light-responsive, potent modulator of cellular circadian rhythms. Remarkably, due to its thermal and metabolic stability, it was possible to employ compound **9** as ON but also OFF photochemical modulator of the circadian period, depending on which isomer is initially present in the cells. Together with the successful use of visible light for in situ photoisomerization, these results pave the way for future in vivo studies where the circadian

period can be modulated reversibly with the prospect of high spatiotemporal resolution.

## Methods

**Organic synthesis**. All chemicals were purchased from Sigma–Aldrich, Acros, Fluka, Fischer, TCI and were used as received. MBraun SPS-800 solvent purification system was used to obtain dry DCM, and aqueous solutions were prepared with deionized water. TLC analysis was performed on F254 silica gel plates with fluorescence-indicator UV254 (Merck, Kieselgel 60) and the UV-active compounds were detected using UV light (254 nm or 365 nm). For the additional visualization, oxidative staining of TLC plates with aqueous basic potassium permanganate solution ($KMnO_4$) or aqueous acidic cerium phosphomolybdic acid solution (Seebach's stain) was used. Organic solutions were dried over $MgSO_4$ and solvents were removed in *vacuo* (Büchi, R-300).

**Analytical procedures**. $^1$H NMR, $^{13}$C NMR, and $^{19}$F NMR spectra were recorded at room temperature (22–24 °C) on a 400-MHz Agilent Technologies 400-MR (400/54 Premium Shielded) spectrometer. As an internal reference, residual solvent peaks were used [CDCl$_3$: $\delta_H$ = 7.26 ppm; CDCl$_3$: $\delta_C$ = 77.16 ppm; DMSO-$d_6$: $\delta_H$ = 2.50 ppm; DMSO-$d_6$: $\delta_C$ = 39.52 ppm]. The multiplicities of the signals are denoted by s (singlet), d (doublet), t (triplet), q (quartet), hept (heptet), m (multiplet), and br (broad signal). High-resolution mass spectrometric measurements were performed using a Thermo scientific LTQ OrbitrapXL spectrometer with ESI ionization. Melting points were recorded using a Stuart analog capillary melting point SMP11 apparatus. All microwave reactions were performed in CEM Discover SP-D microwave reactor. UV-Vis absorption spectra were recorded on an Agilent 8453 UV-Visible Spectrophotometer.

**In vitro kinase assay**. A stock solution of 7.4 mM in DMSO was prepared and thermally adapted by heating to c.a. 100 °C for 1 min. Two series of eight samples were prepared with consecutive three-time dilutions from the stock solution. One of the series ('dark' samples) was prepared in dark Eppendorf tube, the other series ('light' samples) was pre-irradiated with UV light (365 nm, UV lamp Spectroline, ENB-280/FE, 1 × 8 Watt) for 45 min. For compound **9**, 1 mM solution in DMSO was thermally adapted by heating to 80 °C for 2 min and split into two vials. One vial was kept in the dark and the other was irradiated with green light (530 nm, 3x Nichia NCSG219B-V1, 3 × 550 mW, Sahlmann Photochemical Solutions) for 60 min. Both dark-kept and irradiated samples were subjected to 8-point 3-fold

dilutions. All the solutions were applied to the well plate under a red lamp in the dark room in order to prevent ambient-light-induced photo-isomerization.

The assays were performed on a white, solid-bottom 384-well plates. The total volume for the reaction is 10.5 μl. Firstly, a solution of CKI and peptide was added to the bottom of the well (9 μL). Next, a corresponding solution of the compound (0.5 μL, final 5% DMSO) followed by 50 μM ATP solution (1 μL) were pipetted into the upper corners of each well. The enzymatic reaction was started by spinning down the plate (1800 x g, 2 min).

By employing this method, all reactions were started at the same time, minimizing variance between different samples. Incubation for 3 h at 30 °C or 2 h at 37 °C allowed for the enzymatic phosphorylation of the substrate peptide. As the reaction started, the wells with 'light' samples were irradiated with UV light or kept in dark in case of experiment with compound **9**. 'Dark' (0 min pre-irradiation) wells were covered with an aluminum sticker from the beginning. After the incubation, 10 μL Kinase Glo® (Promega) was applied into the wells and the luminescent signal was recorded by a plate reader (BioTek Synergy H1 or Cytation).

**Cellular assays.** Stable U2OS reporter cells harboring *Bmal1-dLuc* reporter were suspended in phenol red-free culture medium [DMEM (D2902, Sigma) supplemented with 10% fetal bovine serum, 3.5 mg/mL D-glucose, 3.7 mg/mL sodium bicarbonate, 0.29 mg/mL L-glutamine, 100 units/mL penicillin, and 100 μg/mL streptomycin] and plated onto a white, solid-bottom 384-well plates at 30 μL (3000 cells) per well. After 2 days, 40 μl of phenol red-free explant medium [DMEM (D2902, Sigma) supplemented with 2% B27 (Gibco), 10 mM HEPES, 3.5 mg/mL D-glucose, 0.38 mg/mL sodium bicarbonate, 0.29 mg/mL L-glutamine, 100 units/mL penicillin, 100 μg/mL streptomycin, and 0.2 mM luciferin; pH 7.2] was dispensed into each well, followed by the application of 0.5 μL of compounds (dissolved in DMSO; final 0.7% DMSO; pre-irradiated with UV for 60 min, white LED for 10 min, and/or green LED for 60 min when indicated). The plate was covered with an optically clear film, and subjected to irradiation with 530 nm LED lamp (3 × 550 mW, λ_max = 530 nm, FWHM 36 nm, Sahlmann Photochemical Solutions) or 400 nm LED lamp (3 × 1000 mW, λ_max = 400 nm, FWHM 11.9 nm, Sahlmann Photochemical Solutions) from 12 cm distance for 30 min. Compound handling and irradiation were conducted in a dark room under red lamp illumination. Luminescence was recorded every 100 min in a microplate reader, Infinite M200Pro (Tecan). For the phase control experiments, cellular light irradiation was performed in a dark 32 °C room and luminescence was recorded in a microplate reader Synergy H1 (BioTek) placed in the same room, to avoid possible effect of transient temperature change on the phase. Circadian period and phase were determined from luminescence rhythms by a curve fitting program MultiCycle (Actimetrics). Data from the first day was excluded from analysis, because of transient changes in luminescence upon medium change. Statistical significance was evaluated using two-way analysis of variance (ANOVA) followed by a Sidak's multiple comparisons test or one-way ANOVA followed by a Tukey's multiple comparisons test using Prism software (GraphPad Software).

**Ex vivo analysis.** For ex vivo analyses, all mouse studies were approved by the Animal Experiment Committee of Nagoya University and performed in accordance with guidelines. *Per2::Luc* knock-in mice were obtained from Dr. Joseph S. Takahashi. They were reared in the animal facilities where environmental conditions were controlled (12-h light and 12-h dark cycles, the ambient temperature at 23 ± 2 °C, and humidity at 60 ± 10%). Spleen tissue was dissected, and tissue pieces were placed in phenol red-free explant medium without luciferin and containing compounds (final 0.18% DMSO) in a black, clear-bottom 24-well plate. The plate was covered with an optically clear film, and subjected to irradiation with LED lamp (λ_max = 530 nm) from 12 cm distance for 30 min or kept in dark. Luciferin (final 1 mM) was supplemented to the medium, and luminescence was recorded every 30 min for 5 days in a LumiCEC luminometer (Churitsu). Circadian period was determined from luminescence rhythms by MultiCycle. Data from the first day was excluded from analysis, because of transient changes in luminescence upon medium change. Statistical significance was evaluated using two-sided Student's *t* test using Prism software.

To harvest the suprachiasmatic nucleus (SCN), *Per2::Luc* heterozygote neonatal mice at postnatal day four to seven were euthanized. Coronal SCN slices of 300 μm thick were made with a tissue chopper (Mcllwain). The SCN tissue was dissected at the mid-rostrocaudal region and a paired SCN was cultured on a Millicell-CM culture insert (Millipore Corporation)[26]. The slice was cultured in 5% CO_2/95% air at 36.5 °C with 1 ml DMEM (Invitrogen) with 5% fetal bovine serum for three or five days, and then the culture medium was exchanged and measurement of Per2::Luc bioluminescence was started by using luminometor (Kronos, Atto). D-Luciferin (final 1 mM) (Wako Pure Chemical) was supplemented to the medium. Compound was applied into the culture medium on day 4 of the measurement. On day 8, the tissue was subjected to irradiation with LED lamp (λ_max = 530 nm) from 11 cm distance. Circadian period was calculated by a cosine curve fitting. Statistical significance was evaluated using one-sided Friedman test with post hoc Steel-Dwass test.

**Molecular docking.** Two crystal structures of CK1α are available in the Protein Data Bank, one is in apo form (PDB ID: 5FQD) and the other in complex with a ligand at the ATP binding site (PDB ID:6GZD). Both of these structures were used for the docking studies of compounds. There were two molecules of CK1α in the asymmetric unit of 5FQD (Chain ids C and F). The conformation of the two chains is identical with RMSD of 0.29 Å. Hence, only one chain (Chain C) was used for docking into the apo form. The asymmetric unit of ligand bound form (PDB ID: 6GZD) contained single molecule, which was used for docking studies.

Prior to docking of the compounds, both the protein structures were prepared using Protein Preparation Wizard of Schrodinger software suite (Schrödinger Release 2018-4: Protein Preparation Wizard; Epik, Schrödinger, LLC, New York, NY, 2016; Impact, Schrödinger, LLC, New York, NY, 2016; Prime, Schrödinger, LLC, New York, NY, 2018). This involved addition of missing side chain atoms, addition of hydrogens, optimization of hydrogen bonds and restrained minimization of the protein molecules with convergence to maximum RMSD of 0.3 Å.

The *trans* and *cis* forms of each molecule were subjected to ligand preparation protocol of Schrodinger using LigPrep (Schrödinger Release 2018-4: LigPrep, Schrödinger, LLC, New York, NY, 2018). The prepared ligands were then docked to the ATP binding site using Glide XP (Schrödinger Release 2018-4: Glide, Schrödinger, LLC, New York, NY, 2018)[27]. After each docking run, the protein-ligand complexes with at least one hydrogen bond to the hinge region, more specifically to the backbone of L93 residue of CK1α and high glide score were selected for further analysis.

Molecular dynamics simulation was performed in explicit solvent for each of the selected complexes for 1.2 ns using Desmond[28] (Schrödinger Release 2018-4: Desmond Molecular Dynamics System, D. E. Shaw Research, New York, NY, 2018. Maestro-Desmond Interoperability Tools, Schrödinger, New York, NY, 2018) and OPLS3e force field[29]. The protein-ligand complex was solvated in an octahedral box with TIP3P water molecules. L-BFGS energy minimization was performed for the whole system with convergence threshold of 1 kcal/mol/Å. The energy minimized system was then equilibrated in an NVT ensemble at 10 K for 100 ps using Brownian Dynamics and restraints on solute heavy atoms. This was followed by NVT equilibration at 10 K for 12 ps with restraints on solute heavy atoms. A 12-ps NPT equilibration was then performed at 10 K with restraints on solute heavy atoms. Further equilibration at 310 K in NPT ensemble was performed for 12 ps and 24 ps with and without restraints on solute heavy atoms respectively. The equilibrated system was used to perform molecular dynamics simulation in NPT conditions at 310 K for 1.2 ns without any restraints. The MD trajectories were analyzed with the "simulation quality analysis" in Desmond and the last 1 ns trajectory was used for further analysis.

Twenty equally spaced frames were extracted from the last 1 ns trajectory of each protein-ligand complex and MM-GBSA calculations were performed for each frame using Prime module of Schrodinger (Schrödinger Release 2018-4: Prime, Schrödinger, LLC, New York, NY, 2018).

**Reporting summary.** Further information on research design is available in the Nature Research Reporting Summary linked to this article.

## Data availability
The authors declare that the data supporting the findings of this study are available within the paper and its supplementary information files. Additional data on methods used are available from the corresponding author upon reasonable request.

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

## Acknowledgements

We thank Dr. Kaori Goto for technical assistance and Dr. Joseph S. Takahashi for *Per2::Luc* knockin mice. We gratefully acknowledge generous support from The Netherlands Organization for Scientific Research (NWO-CW, Top grant to B.L.F., and VIDI Grant No. 723.014.001 for W.S.), the Royal Netherlands Academy of Arts and Sciences Science (KNAW), the Ministry of Education, Culture and Science (Gravitation program 024.001.035), the European Research Council (Advanced Investigator Grant No. 227897 to B.L.F.), Grant-in-Aid for Scientific Research (B) 18H02402 and Challenging Research (Exploratory) 18K19171 and 20K21269 from JSPS (T.H.), Uehara Memorial Foundation (T.H) Takeda Science Foundation (T.H.), Ichiro Kanehara Foundation for the Promotion of Medical Sciences and Medical Care (T.H.), Grant-in-Aid 18H02477 from JSPS (D. O.), and SECOM Science and Technology Foundation (D.O.). D.K. acknowledges the receipt of a fellowship from the Dositeja Fund for Young Talents for international studies, thanks Dr. Nadja Simeth for an insightful synthetic suggestion for compound **8** and Dr. Kaja Sitkowska for drawing light bulbs.

## Author contributions

B.L.F., T.H., and W.S. guided the research; D.K. and B.L.F. designed photoswitchable modulators and together with T.H., W.S., K.I., and C.M.V. designed the experiments; D.K. and C.M.V. synthesized photoswitchable analogs of longdaysin, performed photochemical studies, and in vitro assays; A.Su., D.O., Y.N., M.I., and T.H. performed and analyzed in vitro, cellular, and tissue experiments; A.Sr. and F.T. designed and ran docking simulations; D.K., T.H., and B.L.F. wrote the manuscript with support and contributions from all authors.

## Competing interests

The authors declare no competing interests.
