## [Peer Review File · Nature Communications]

Reviewer Comments:

Reviewer #1:

Feringa, Hirota, Szymanski and colleagues describe a photoswitchable version of longdaysin, an inhibitor of Casein kinase 1 epsilon, which has a profound effect on the period of the circadian rhythm. This study mirrors their previous work on a photocaged version of longdaysin but uses a reversible photoswitch instead of a caging group that can be used only once.

The article provides an instructive template for the rational development of photoswitchable inhibitors that are marked by favorable photostationary states, metabolic stability, significant changes in bioactivity upon photoisomerization, and sensitivity toward visible light. The authors ultimately arrive at compound 9, which incorporates the tetrafluoro azobenzenes pioneered by Hecht. This photoswitch shows increased thermal bi-stability, and is surprisingly stable metabolically despite their electron deficient nature. However, the optimization only leads to a compound with moderate differences between cis (17.3 μM) and trans (4.6 μM).

I cannot see how this new tool compound is significantly better than the caged version or how it will be widely used by circadian rhythm community. What new insights can be gained that inform the biology or what concrete therapeutic applications can be envisioned? That the circadian rhythm can be influenced with a photochemical approach has been demonstrated before by the same group before.

The selectivity observed between the different casein kinase 1 isoforms is interesting and could, as the authors point out themselves, lead to new discoveries regarding the relative importance of the isoforms. However, they do not provide any new insights or discoveries in this regard. If this were the case, the paper would be significantly stronger.

Several control experiments are needed that demonstrate that light itself does not affect the circadian rhythm. These include the use of luciferin only and vehicle control in the presence and absence of irradiation. In addition the authors should in the figure caption what the nature of the error bars is, and include statistical analysis for the bar diagrams in Figure 4.

If the authors include the necessary controls and are able to design a more elaborate biological application, e.g. *in vivo*, this paper may become suitable for publication in *Nat. Comm.* In its current form the paper describes the genesis of the optimal photoswitch at length, is somewhat technical, and is better suited for a more specialized journal with an emphasis on chemical biology and photochemistry.

Reviewer #2:

This manuscript by Kolarski and co-workers is quite exciting and well worthy of publication in Nature Communications, after the authors have addressed the following issues. There is currently nothing wrong in what is in the manuscript, but it unfortunately finishes prematurely.

Major comments:

Where are the raw data, i.e. the luminescence traces corresponding to figure 4? The very tiny oscillations at the bottom of Fig 4 are only graphical representations of what the data should, or is expected to, look like, right? However, to truly appreciate the reversibility of this small molecule, raw luminescence data showing irradiation-dependent temporary changes in period followed by recovery are absolutely required, even if it is as supplementary data.

The second major comment is linked to the first. I believe that the true power of such a photoswitch is the ability to elicit not only circadian period changes, but also phase changes. When one thinks about the relationship between circadian period and phase, a temporary change in period will also lead to a phase change. Can the authors perform in vitro experiments in which a jetlag-like state, i.e. a difference in phase between two cell populations, for example caused by initial entrainment to different temperature cycles, can be corrected if cells loaded with compound cis-9 by irradiating the cells with the appropriate wavelength to cause temporary conversion to trans-9 and associated period change, then reverting to cis-9 by irradiating the cells with the appropriate wavelength again once it has reached the desired phase (and back to the "normal period", same as the other group of cells who were not drugged). This would really show the power of a pharmacological photoswitch, and may even provide a proof of concept for new photochemotherapy of circadian misalignments.

Minor comment is only that units for luminescence have not been specified (4e for example).

Jean-Michel Fustin

Reviewer #3:

The work presented by D. Kolarski et al. introduces the development of 7 photoswitchable inhibitors of casein kinase I (CKI). The best performing compound 9 allows for the first time to reversibly slow down cell-autonomous circadian rhythms (CRs) in vitro by the means of light. The photocontrolled modulators of circadian rhythm have been designed by azo-extension of the chemical structure of the purine-based CKI inhibitor longdaysin. This drug design permits an important photopharmacological evolution of the previous non-reversible "photocleavable" approach for light-controlling the CRs that was described by the same authors in 2019 (<https://doi.org/10.1021/jacs.9b05445>).

This work of photopharmacology is outstanding from different points of view:

* The medicinal chemistry work is very robust and elegant: the authors developed 7 azobenzene-based derivatives of longdaysin starting from the identification of the best aromatic position for extending the purine base core with a simple "unsubstituted" azobenzene in order to obtain the photocontrolled inhibition of CKI. Once the positioning of the purine base was found optimal at "para" with respect to the azo bridge, the authors gradually improved the photochemical properties of the inhibitors by introducing different chemical substituents at the azobenzene moiety until they obtained a photoswitchable compound (9) that presents the suitable photochemical behavior for such biological application: sufficient photochemical stability for both the isomers in the aqueous medium (> 50 hours), a negligible photo-fatigue, and photoisomerization in the visible range of wavelengths.

* The pharmacological activity of all the synthesized compounds was fully characterized in vitro in

human cells, where the inhibition of CKI, and the corresponding effect on the CRs, was studied for both the trans and cis-enriched forms.

* The syntheses, the chemical/photochemical characterizations, and the studies of the compounds' stability in biological mimicking conditions that are reported in the main text and in the SI are very robust and complete for all the compounds, and should allow the reproduction of the chemical results that are described in the main text and in the SI.

* The biological results that have been obtained are original and outstanding in the fields of (photo)pharmacology. This work opens the door to the potential photopharmacological control of CRs in more complex systems using compound 9 and visible light, which is more suitable for applications in wildtype animal models.

* Moreover, the authors demonstrate that the photoswitchable behavior of compound 9 mostly affects the CKI δ isoform, whereas it is not evident for CKI α , thus accounting for the biological results obtained in cells and allowing a very interesting discussion about the different functions of these class of proteins in the regulation of the CRs.

Thus, the manuscript is suitable for publication with the following minor revisions:

* In order to obtain a photochromic analog of longdaysin, the first design approach that comes to mind is azologization rather than azo-extension. Can the authors discuss the reasons for choosing the latter? Perhaps this is intended "to minimize structural and electronic changes introduced" (line 90) but it should be stated more explicitly. In particular, molecular docking calculations like those shown in the SI might tell if non-canonical azologization of the longdaysin chemical structure (i.e. replacing with an azo-bridge the aminomethylenic bridge between the original aromatic ring and the aromatic purine base moiety) can lead to active, photoswitchable inhibitors of CKI.

* The main text is very well written but might be shortened to improve readability in this article format.

* In Figure 1a: gene expression / time graph, it could be indicated what gray lines correspond to (basal gene expression in "normal" conditions?)

* In line 201 it is stated that "recovery of the biological activity was achieved upon back-isomerization with white light (red lines, Figure 3c)." However, the recovery of the biological activity that was achieved upon back-isomerization with white light is complete for compound 8, but only partial for compound 7.

* Figure 4: The authors used blue color to indicate 400 nm light, which is actually violet. It might be changed to avoid misunderstanding during the reading.

* In Figure 2 caption, line 112, "dillution" is misspelled.

Response to the comments of Reviewer 1:

1.1. *The article provides an instructive template for the rational development of photoswitchable inhibitors that are marked by favorable photostationary states, metabolic stability, significant changes in bioactivity upon photoisomerization, and sensitivity toward visible light. The authors ultimately arrive at compound 9, which incorporates the tetrafluoro azobenzenes pioneered by Hecht. These photoswitch shows increased thermal bistability, and is surprisingly stable metabolically despite their electron deficient nature. However, the optimization only leads to a compound with moderate differences between cis (17.3 μM) and trans (4.6 μM). I cannot see how this new tool compound is significantly better than the caged version or how it will be widely used by circadian rhythm community. What new insights can be gained that inform the biology or what concrete therapeutic applications can be envisioned? That the circadian rhythm can be influenced with a photochemical approach has been demonstrated before by the same group before.*

RESPONSE: Photoswitchable modulators of circadian rhythms, described in this work, comprise a significant improvement in comparison to the caged compounds due to the reversible nature of their activation with light. Caged compounds, once activated, cannot undergo deactivation. This severely limits their applications due to typically long-term response in circadian biology: diffusion would lead to reduced temporal and spatial control of the drugs' activity, therefore making spatiotemporal fine-tuning of the circadian rhythm difficult. The use of reversibly activated compounds allows their inactivation with light around the intended time and site of action, leading to the limited exposure of other parts to the active form. Although the activity difference between *cis* and *trans* isomers against CKI enzyme was moderate, they showed significant difference of circadian period change. Controlling circadian rhythm in a fully reversible manner is unique and totally unprecedented. Furthermore, in view of the long-time response, we have beyond doubt push the frontiers of photopharmacology. We demonstrate now that the concept works in cells and tissues, providing an excellent basis for future studies in several directions. Therefore, our photoswitchable modulators provide a powerful tool for future application to the spatiotemporal control of the circadian clock.

The key questions that chronobiology aims to address include communication among cellular circadian clocks. Local and reversible control of the cellular clock enabled by photoswitchable modulators will open the door to address challenging issues regarding cellular communications. Another potential application of photoswitchable modulators is the circadian phase control as suggested by Reviewer 2. Now we provide the key experiment demonstrating that a transient change of the period achieved by the reversible system resulted in the circadian phase shift (new Figure 7), paving the way for tackling the adjustment of the phase shift caused under jet lag conditions. Altogether, we see a significant potential of using the reversible system to answer numerous questions regarding spatiotemporal resolution.

1.2. *The selectivity observe between the different casein kinase 1 isoforms is interesting and could, as the authors point of themselves, lead to new discoveries regarding the relative importance of the isoforms. However, they authors provide any new insights or discoveries I this regard. If this were the case, the paper would be significantly stronger.*

RESPONSE: We would like to thank Reviewer 1 for the comment and suggestion. The CKI isoform selectivity is an exciting feature of photoswitchable modulator **9**. However, obtaining mechanistic insight is challenging due to the almost identical binding pocket residues of CK1 α and CK1 δ . To approach this selectivity aspect, we have conducted docking studies of *cis*- and *trans*-**9** to both isoforms. Although small differences in the IC₅₀ values are difficult to reproduce in docking and molecular mechanics-based binding energy calculations, we have observed qualitative differences in binding (new Figure S38b) that are in agreement with the experimental results (Figure 4d). There is an additional stacking interaction with the C-terminal F295 in case of CK1 δ -*trans*-**9** complex, which could be contributing to the lower binding energy. The corresponding residue in CK1 α is Q303 and quite far from the binding pocket.

The following explanation has been added to page 12, at the end of the first paragraph and it reads:

*To further explain the difference in the CKI-isoform affinity, we used molecular docking to investigate and compare the binding energy of *trans*-**9** and *cis*-**9** in complex with CK1 δ with that of CK1 α (Figure S38b). The*

binding energy for trans-9 was lower than for cis-9 in case of CK1 δ complex (Figure S38b), corroborating the IC₅₀ values determined for these compounds in vitro (Figure 4d).

1.3. Several control experiments are needed that demonstrate that light itself does not affect the circadian rhythm. These include the use of luciferin only and vehicle control in the presence and absence of irradiation. In addition the authors should in the figure caption what the nature of the error bars is, and include statistical analysis for the bar diagrams in Figure 4.

RESPONSE: Some of the control experiments were not shown in the original manuscript, although we had the data. Now all control data are included in the supplementary data (Figures S24-29). Statistical analysis and the explanation of the error bars have been added in the revised manuscript.

1.4. If the authors include the necessary controls and are able to design a more elaborate biological application, e.g. in vivo, this paper may become suitable for publication in Nat. Comm. In its current form the paper describes the genesis of the optimal photoswitch at length, is somewhat technical, and is better suited for a more specialized journal with an emphasis on chemical biology and photochemistry.

RESPONSE: We would like to thank again Reviewer 1 for all the comments. We have conducted *ex vivo* studies using peripheral spleen tissue as well as the central suprachiasmatic nucleus that controls behavioural rhythms. We see significant decrease of period-lengthening effect of the compound by green light irradiation (new Figure 6). These results provide an important step forward for future *in vivo* studies.

With all original and newly added data, we consider that our approach is not only technical, since it is an interdisciplinary effort that presents deeper understanding of all the fields involved, and establishes all the basic features, as well as shows the first time the reversible control of a long-term biological process. Together with the ability to modulate crucial circadian parameters in cells and tissues, this manuscript shows development, analysis, understanding and application of photopharmacology in chronobiology, making it unique in both fields.

Response to the comments of Reviewer 2:

2.1. *Where are the raw data, i.e. the luminescence traces corresponding to figure 4? The very tiny oscillations at the bottom of Fig 4 are only graphical representations of what the data should, or is expected to, look like, right? However, to truly appreciate the reversibility of this small molecule, raw luminescence data showing irradiation-dependent temporary changes in period followed by recovery are absolutely required, even if it is as supplementary data.*

RESPONSE: Some of the raw luminescence traces were missing in the original manuscript. All traces are included in the revised manuscript.

2.2. *The second major comment is linked to the first. I believe that the true power of such a photoswitch is the ability to elicit not only circadian period changes, but also phase changes. When one thinks about the relationship between circadian period and phase, a temporary change in period will also lead to a phase change. Can the authors perform in vitro experiments in which a jetlag-like state, i.e. a difference in phase between two cell populations, for example caused by initial entrainment to different temperature cycles, can be corrected is cells loaded with compound cis-9 by irradiating the cells with the appropriate wavelength to cause temporary conversion to trans-9 and associated period change, then reverting to cis-9 by irradiating the cells with the appropriate wavelength again once it has reached the desired phase (and back to the "normal period", same as the other group of cells who were not drugged). This would really show the power of a pharmacological photoswitch, and may even provide a proof of concept for new photochemotherapy of circadian misalignments.*

RESPONSE: We would like to thank Reviewer 2 for this very insightful suggestion, which prompted us to perform the key experiment shown in new Figure 7 of the revised manuscript. As suggested by the reviewer, this reversible manipulation led to a successful phase shift by transient period change, providing the proof of concept for the newly developed reversible system in addressing circadian misalignments.

2.3 *Minor comment is only that units for luminescence have not been specified (4e for example).*

RESPONSE: It is arbitrary unit. We added explanation in figure legends.

Response to the comments of Reviewer 3:

3.1. *In order to obtain a photochromic analog of longdaysin, the first design approach that comes to mind is azologization rather than azo-extension. Can the authors discuss the reasons for choosing the latter? Perhaps this is intended “to minimize structural and electronic changes introduced” (line 90) but it should be stated more explicitly. In particular, molecular docking calculations like those shown in the SI might tell if non-canonical azologization of the longdaysin chemical structure (i.e. replacing with an azo-bridge the aminomethylene bridge between the original aromatic ring and the aromatic purine base moiety) can lead to active, photoswitchable inhibitors of CKI.*

RESPONSE: We agree with Reviewer 3 that replacing aminomethylene bridge of longdaysin looks like a promising and straightforward position for rendering longdaysin photo-responsive. In fact, we have initially performed this transformation, but the obtained photoswitches showed undesired photochemical properties (short half-life and low PSS distributions). Also, those compounds were susceptible to diazo bond reduction, yielding corresponding hydrazines and preventing photomodulation of the circadian rhythm (to be published elsewhere). This data revealed key challenges for long-term modulation of biological activity and prompted us to explore the current structural approach.

3.2. *The main text is very well written but might be shortened to improve readability in this article format.*

RESPONSE: We appreciate the suggestion from Reviewer 3. However, the article length is shorter than the recommended 5000 words, even after additions made at the current revision stage. Having in mind multiple topics discussed and comprehensive chemical, photochemical and biological studies performed, we would like to keep it in the form as it is now. If it comes to be crucial for the publication, we will gladly reduce the text.

3.3. *In Figure 1a: gene expression / time graph, it could be indicated what gray lines correspond to (basal gene expression in “normal” conditions?)*

RESPONSE: We agree with Reviewer 3 that legend could help better understanding. Thus, it was added within Figure 1 of the revised manuscript.

3.4 *In line 201 it is stated that “recovery of the biological activity was achieved upon back-isomerization with white light (red lines, Figure 3c).” However, the recovery of the biological activity that was achieved upon back-isomerization with white light is complete for compound 8, but only partial for compound 7.*

RESPONSE: In the revised version, it has been changed to: *Furthermore, a partial recovery of the biological activity for 7 and full recovery in case of 8 was achieved upon back-isomerization with white light (red lines, Figure 3c).*

3.5 *Figure 4: The authors used blue color to indicate 400 nm light, which is actually violet. It might be changed to avoid misunderstanding during the reading.*

RESPONSE: The color has been adjusted to violet in the revised version of the manuscript.

3.6. *In Figure 2 caption, line 112, “dillution” is misspelled.*

RESPONSE: It is corrected to ‘dilution’ in the revised version of the manuscript.

REVIEWERS' COMMENTS:

Reviewer #2 (Remarks to the Author):

The authors have addressed the comments of all reviewers appropriately, and the addition of new data significantly increases the impact and relevance of this study. It is my opinion that this manuscript should now be prepared for publication.

Jean-Michel Fustin

Reviewer #3 (Remarks to the Author):

All the comments have been addressed, the manuscript is suitable for publication